# The Community Collaboration Model for School Improvement: A Scoping Review

**Dawn Anderson-Butcher** [1,*]**, Samantha Bates** [1] 🄳**, Hal A. Lawson** [2]**, Tasha M. Childs** [3] 🄳 **and Aidyn L. Iachini** [3]

1 College of Social Work, The Ohio State University, Columbus, OH 43210, USA
2 School of Social Welfare, University at Albany, Albany, NY 12222, USA
3 College of Social Work, University of South Carolina, Columbia, SC 29208, USA
* Correspondence: anderson-butcher.1@osu.edu

**Abstract:** Schools worldwide are developing innovative models in response to, and in anticipation of, societal changes. Aiming to address non-academic barriers to learning, while capitalizing on out-of-school time, some school and community leaders have prioritized family and community partnerships, especially in the United States (U.S.). The Community Collaboration Model (CCM) is one such U.S. partnership-oriented model of expanded school improvement. In contrast to some partnership-oriented models, the CCM prioritizes improvements in classrooms and communities, aiming to support students, assist teachers, and improve relationships beyond the typical school day. This scoping review examines 14 peer-reviewed articles which describe CCM-centered innovations and documented outcomes. Barriers and facilitators associated with CCM adoption and implementation in diverse U.S. school and community settings also are explored. CCM's contributions to important student and school outcomes (e.g., increased access to mental health services, improved school climate, decreased discipline referrals) are documented in this scoping review. Some researchers have also described implementation-related facilitators (e.g., partnerships with universities) and barriers (e.g., initial resistance by educators) that influence the utility of the model in practice. Drawing on prior research, the authors discuss findings and implications for future research, educational policy, and practice.

**Keywords:** expanded school improvement; non-academic barriers; school-family-community partnerships; out-of-school time; scoping review





## 1. Introduction

School improvement policies in education systems aim to support student learning and ensure young people are prepared to join today's workforce and global economy. In the United States (U.S.), school improvement is a formative and essential topic guiding current educational policy, school funding decisions, and academic debates. Indeed, educational scholars in the U.S. have systemically studied school improvement processes since the early 1940s by first focusing on the diffusion of products into curricula and then shifting toward the exploration of social and organizational aspects of institutional change [1]. This lens has and continues to guide policy and practice in the U.S., United Kingdom, and Australia, which all recognize school improvements as key initiatives on their national policy agendas. Ultimately, accountability tied to school funding has only increased the need to understand whether these policies contribute to student success in the U.S. [2,3]. Beyond the U.S., organizations such as the United Nations Education, Scientific, and Cultural Organization [2] also have a vested interest in understanding, documenting, and disseminating justifiable school improvement policies—with special attention to the translation of research-supported, optimal practices into classrooms, and schools around the world. While our study is contextualized to school improvement in the U.S., it is important to note that policy leaders, governmental officials, educators, and human rights

organizations from across the globe have a vested interest in this topic. All see access to quality education and effective approaches to school improvement as mechanisms that can support global efforts to eradicate poverty, prevent social exclusion, improve public health, and facilitate equitable, sustainable economic growth [2,3].

An important distinction is foundational to these national and international innovations. Research directed at school improvement models and strategies in the U.S. is conceptually different from research on school effectiveness. While school effectiveness research focuses on identifiable student and school performance-based outcomes [3], school improvement policies and research focus on salient innovations and progress markers such as attendance, suspensions and expulsions, and retention-turnover rates for students and educators, with a special interest in the interventions that make a positive difference as well as school decision-making networks and broader leadership structures.

Because U.S. schools and their host community contexts vary, as do schools' implementation of different school improvement models and strategies, researchers and evaluators confront endemic, formidable complexity. Indeed, scholars in the U.S. and worldwide argue the intersection between school improvement policy and practice is not well understood, and some lament the lack of sufficient, comparative research which explores and explains "what works" in different contexts [3,4].

The quest for innovative, theoretically-sound, evidence-based, and data-driven school improvement models also derives from a potent combination of demographic, technological, and societal changes. Unprecedented cross-border migration of the world's children and families includes the challenges and opportunities associated with ethnic, cultural, and linguistic diversity [5]. Predictably, the potent combination of poverty, social exclusion, and social isolation is concentrated in specific locales and impacts local schools and community service agencies [5,6].

Where schools are concerned, educators confront challenges categorized as "nonacademic barriers to learning", and more fundamentally, barriers to attendance, on-time arrival, and student engagement. Barriers include food insecurity, homelessness, family violence, mental health concerns, and even hopelessness. These vulnerabilities and others they implicate increase risks for sub-optimal school performance and adverse developmental outcomes (e.g., school dropout, substance use and misuse, and incarceration).

The international scale of these challenges frames an important call to action within education. For example, poverty rates among school-aged students around the world are alarming. An estimated 700 million students and their families experience economic insecurity each year [2]. Globally, mental health and behavioral disorders, suicide, and youth violence also are of growing concern, especially among our adolescent population [7,8]. Indeed, recent data from the Center for Disease Control and Prevention [9] in the United States suggest a 40% increase in anxiety, depression, and thoughts of self-harm among adolescents over the past decade. Furthermore, nearly one-fifth of students in the United States are either living in poverty, attending a high-poverty school, or both [10]. Global data further demonstrate the need, as one out of every five children and adolescents is estimated to have some sort of social-emotional disturbance, a rate that has more than doubled in the last five years [11]. Where educators and schools are concerned, all such child and family-related needs tend to be classified as "nonacademic barriers to learning". Alone and together, these barriers influence schools' outcomes. For example, students in schools in identifiable places challenged by poverty, social exclusion, and social isolation worldwide struggle to read at grade level, develop skills to enter the workforce, secure a living wage as adult citizens, and contribute to local community development [2,12].

The idea of a stand-alone school focused primarily on academic learning and achievement, a model in which educators, particularly teachers, work alone, no longer is tenable or effective in the U.S. [12]. In response, government officials, researchers, policymakers, school leaders, and educators have designed and implemented expanded partnership models to address the various factors and forces that impact students' attendance, engagement, learning, healthy development, academic achievement, and school completion.

Alongside a renewed focus on academic learning and achievement, leaders have shifted their attention to addressing and preventing an array of adverse influences typically called "non-academic barriers" to school success. This new, broader focus has taken educators and policymakers outside the school's walls and beyond the school day, necessitating partnerships with community agencies, families, higher education institutions, and vocational training organizations.

## 1.1. Partnership Models

Several national and international partnership models help schools address the forces and factors that influence student learning [13]. Examples in the U.S. include Full-Service Community Schools [14], comprehensive systems of learning supports [15], the coordinated school health model [16], and the Whole School, Whole Community, Whole Child approach [17–19]. Other international models exist, such as full-service extended schools in the United Kingdom, which also make linkages to social services to support service access, adult learning, and academic interventions. All such partnership models require educators to work closely with community leaders and service providers, maximizing their respective assets, resources, and services (especially in relation to access to services such as health, mental health, and services to meet basic needs) and parents/caregivers to improve school conditions [20]. Significantly, these partnership models focus on more than academic learning and school performance. Models utilize locally driven strategies to integrate student supports, expand, and enrich learning time and opportunities, engage families and the community, and foster collaborative leadership [21]. Models demonstrate the value of collaborative partnerships across systems to guide school planning and change efforts.

Several U.S. partnership models, such as City Connects [22], Communities In Schools [23,24], and the National Youth Advocate Program [25], also frame how schools can cultivate partnerships with community agencies and institutions, and leverage school-based practitioners to address nonacademic learning barriers and help families navigate social service systems. These models utilize data to identify students' strengths and needs and utilize health and social service personnel (i.e., counselors, case workers, social workers, etc.) to address students' academic, social, emotional, behavioral, and health needs. When examined individually, these partnership models seek to connect students to interventions inside and outside of the school building. Many of these models, however, function as add-on social service delivery approaches that fail to systematically change the way schools are owned and operated and address broader needs that extend into the community. Thus, the exploration of models that aim to bridge schools and communities is needed to further expand school improvement research in the United States and across the globe.

## 1.2. Schoolwide Systems Change

The promise of school improvement models that yield desirable, sustainable outcomes, especially for schools serving socially vulnerable students, is demonstrated in models impacting policy and school operations. The Finnish model is one example. Here, public policy supports high-quality and subsidized child care and early childhood education, student-centered learning focused on mastery and growth, safe and welcoming school cultures, services to address basic needs (i.e., food assistance), learning needs (i.e., counselors, social workers), learning spaces (i.e., libraries, gymnasiums), and other assets [26]. Unfortunately, most policy agendas are not as comprehensive as Finland's model. Indeed, failure to design local and specific school improvement plans often contribute to inefficiencies in planning and implementation, especially given education is a state issue in the U.S. [18,20]. In an analysis of 46 school improvement plans in one U.S. state, scholars identified the ineffectiveness of traditional school improvement processes in mitigating external pressures, implementing instructional shifts, and developing school-wide agreement on goals [27]. The focus of these school improvement plans on academics alone often fails to address underlying structural and community forces that perpetuate inequality and

continue to sustain vast nonacademic needs among students and families, such as mental health concerns, housing instability, and food insecurity [18,27].

Furthermore, traditional school improvement strategies do not account for structures that engage schools in continual improvement and change, such as collecting data to guide school improvement efforts, re-assessing change using annual planning efforts, examining the allocation of school and community resources, or helping schools deploy improvement strategies that create change across multiple systems including peer relationships, family engagement, and community partnerships [28]. Without flexible and sustainable implementation practices that accompany expanded school improvement models, schools struggle to engage in continuous planning and progress monitoring; practices requisite for long-term social change.

### 1.3. Expanded Models of School Improvement

The best models of school improvement comprise efforts to transcend classrooms, while also leveraging the support, resources, and services from non-profit organizations, private businesses, parents and families, government programs, and youth development entities. Through expanded school improvement processes, schools recognize the need to build linkages to family and community resources and to increase youth development opportunities that connect youth to positive environments during out-of-school time [28]. Health and social services are put in place to address non-academic barriers impacting children, youth, and families, and to create learning support systems to link services and interventions back to classrooms to support teachers. Other strategies transform the school climate and culture, through school-wide efforts to support behavior, promote safety, and promote relationships and belonging. They also maximize both school- and community resources for healthy development and learning, creating synergistic investments in children, youth, and families across systems that are needed during this time of increased accountability in education.

### 1.4. Community Collaboration Model (CCM) for School Improvement

The Community Collaboration Model (CCM) is one expanded school improvement model that integrates concepts of positive youth development (PYD) and school-family-community partnership models into one framework. The CCM is similar to existing models of expanded school improvement as it aims to identify supports across multiple sectors to ensure schools meet the holistic needs of students [29–31]. However, the CCM is unique from other expanded school improvement and partnership models because the framework focuses on specific implementation steps such as elevating the voices of multiple school stakeholders to drive improvement plans, mapping resources across five specific pathways, ensuring a comprehensive approach to school reform and partnership development, and providing specific implementation steps to assist schools and their partners through the improvement process. The five pathways guiding the CCM ensure schools look beyond the academic domain and engage in holistic planning based on individual needs, as well as environmental risks and protective factors impacting schools and communities. Additionally, these pathways often guide schools through the successful implementation of the CCM in conjunction and alignment with traditional school improvement processes already underway in educational systems across the globe.

#### 1.4.1. CCM Framework

Theoretically, the CCM framework examines needs and assets across five main pathways: (a) academic learning, (b) youth development and school climate, (c) parent and family engagement, (d) health and social services, and (e) community partnerships [29]. Schools use these pathways to develop strategies and priorities and identify academic and nonacademic barriers to learning. Critical to implementing the CCM is assessing needs across multiple stakeholder groups (i.e., youth, teachers/staff, parents/caregivers, and community members). Figure 1 showcases the theoretical model whereby the CCM begins

with assessing conditions and resources that aim to illuminate assets and gaps across the following five pathways. The model seeks to test whether specific implementation steps grounded across these five pathways improve academic and nonacademic outcomes, and therein strengthen positive developmental outcomes for youth in schools.

## Community Collaboration Model for School Improvement

**Figure 1.** Community Collaboration Model for School Improvement.

### 1.4.2. Academic Learning

The first pathway, academic learning, emphasizes that schools work to examine gaps and align curriculum, instruction, and supports beyond the school day to maximize opportunities for academic learning (i.e., afterschool programs). Central to the CCM is the added focus on extending the reach of academics during out-of-school time. Tutoring, homework help, and other out-of-school supports (i.e., clubs, reading groups, and fluency programs) can increase instructional time and engagement in an academic curriculum [29].

### 1.4.3. PYD and School Climate

The second pathway, youth development, and school climate examine assets and gaps related to school engagement. This pathway focuses on increasing access to opportunities that promote PYD beyond the school day, such as youth sports, positive peer relationships, mentoring, and performance arts. Upon examining youths' assets and needs, CCM strategies implemented through this pathway seek to increase youths' protective factors while mitigating additional risks. For instance, schools examine their implementation of positive behavioral intervention supports (PBIS) to reinforce prosocial behaviors using school-wide and classroom-based interventions [32]. Strategies also aim to improve

the learning environment by focusing on school climate, diversity, and inclusion, and academic motivation [33]. Strategies in this pathway also include an expanded focus on service-learning opportunities, quality afterschool programs, and prevention curricula.

### 1.4.4. Parent/Family Engagement and Support

The third pathway focuses on parent and family engagement and support and seeks to expand upon traditional engagement strategies and support families in their home-school-community environments. Examples include helping parents and caregivers access services to address basic needs, building family resource centers, and supporting parents/caregivers through educational programming, support groups, job training, or additional services. Parent/family engagement beyond efforts to solely involve parents/caregivers in assisting their child with schooling can lead to more buy-in, trust, and positive relationships with families that indirectly impact academic learning.

### 1.4.5. Health and Social Services

The fourth pathway, health, and social services focuses on addressing gaps and leveraging assets related to health, mental health, economic hardships, and other nonacademic barriers. One central goal when addressing needs in this pathway is to increase inter-professional collaboration while relieving some burdens on teachers and school staff. In this model, school leaders are encouraged to examine needs and think about developing collaborative teams charged with addressing specific health, social service, or mental health priorities. Further, school leaders can build partnerships with community health and mental health service providers to meet needs. However, beyond these developments, this pathway helps school leaders and partners reflect on the root causes of problem behaviors and family instabilities. For example, families may face homelessness, food insecurity, community violence, or lack of support to meet their basic needs. Based on barriers identified in this pathway, schools can increase their awareness and empathy regarding family risk factors and improve relationships, linkage mechanisms, and referral systems with community partners to bridge the gap between needs and services.

### 1.4.6. Community Partnerships

The fifth pathway, community partnerships, seeks to improve connections with agencies in the private sector, non-profit organizations, colleges and universities, and other local or state entities. These community partnerships can provide access to additional funding, resources, service-learning opportunities, or school personnel. For example, schools may identify relationships with local universities to create internships that increase access to caring adults and social service supports (i.e., social worker interns, student teachers, etc.). Coordination with community organizations in the private and public sectors also can create single points of contact for families and youth, simplifying the process of receiving services and supports.

### *1.5. CCM Expanded School Improvement Process*

From a practice perspective grounding the CCM in these five pathways can help schools identify expanded priorities that align resources, personnel, and processes to improve conditions for academic and nonacademic success [29]. To guide improvement processes, the CCM is accompanied by key implementation steps that act as a roadmap for schools to engage in continuous and on-going improvement efforts. Key implementation steps include engaging the school and community in a needs/conditions and resources assessment, analyzing gaps, developing new partnerships, programs, and funding streams, strengthening collaborative leadership structures and processes, and evaluating student-, school-, and system-level outcomes. Key implementation steps that guide the CCM are described next.

### 1.5.1. Engaging the School and Community

One of the first steps to adoption is to engage the school and community, fostering readiness and buy-in for an expanded approach to schooling by "building a table" comprised of multiple stakeholders who have a common vision for the work. CCM teams may include leaders from the school, district, community agencies, local government, and families. These stakeholders differ from traditional school teams which often only include internal stakeholders. Extensive time is committed during this milestone to increasing collective understanding of the broadened approach to school-family-community partnerships, as well as fostering commitment to the process and readiness to implement change.

### 1.5.2. Needs/Conditions and Resources Assessment

A second step to adopting the CCM model is to collaboratively assess the current needs of youth and the community using several measures across different stakeholder groups (i.e., youth, teachers, and parents/caregivers) [34]. The process differs from traditional assessments using academic, behavioral, and socioeconomic data from youth which, in turn, fails to consider context and connection to health and social services, youth development avenues, and parent engagement opportunities. In addition to collecting these data, schools also work on teams to map and organize existing resources and programs across the five pathways (i.e., academic learning, health, and mental health services, etc.).

### 1.5.3. Analyzing Gaps

Following data collection and mapping of school resources, school teams can then consider the school's individual and resource needs. Current data, resources, and needs help to identify priorities for improvement and mitigate the duplication of services or supports. For example, a school may realize several supports exist in the youth development and school climate pathway. Still, parent/family engagement supports are needed to strengthen relationships and academic performance outcomes in their school. Notably, these needs may surface through data analysis from different stakeholders. Teacher perceptions, for instance, may show a need for additional supports for youth learning, and comparisons of teacher and parent/caregiver perceptions may indicate gaps in school-family communication methods. Together, teams analyze data and examine resources to strengthen their school environment by addressing needs across systems that influence the youth they serve.

### 1.5.4. Resource/Program Development and Implementation

After analyzing gaps and identifying needs/strengths, the CCM asks improvement teams to ensure their top priorities have a clear place in the improvement process and are interrelated to improving youth's academic and social/emotional outcomes. Schools with few resources may utilize data and mapping processes to realign resources and responsibilities in the school to address gaps across the five pathways. One aim of the model is to ensure schools do not simply add on new programs but instead maximize resources and build partnerships that become an integrated and intentional method to address youth needs and improve outcomes. For example, the process may include integrating evidence-based models, eliminating ineffective models, and implementing culturally responsive approaches in instructional or discipline-related procedures.

### 1.5.5. New Partnerships, Collaborative Leadership, and Infrastructure

Once priorities are set and addressed, school leaders and teams can begin to focus on building partnerships that create effective and lasting relationships in the community or with parents/caregivers. In addition, traditional models of improvement often lack shared accountability for youth success through this collaborative partnership. Through the CCM model, one aim of this process is to strengthen infrastructure in the school by improving or forming data systems, expanding professional development opportunities, changing policy governance, and embedding or restructuring services within each of the five pathways.

In the school, newly expanded school improvement teams can support the linkage and coordination of existing services and designate enhanced roles for current staff to support the new infrastructure of assessment and intervention around youth needs. As partners and leaders work together, instead of competing against each other, schools can implement or maximize existing supports to provide more resources to youth and families.

1.5.6. Evaluating Resources and Programs

A final step in the CCM process includes developing strategies and programs to meet youth's needs inside and outside the school day; however, a critical piece of successful implementation includes evaluating these programs. Evaluation allows schools to examine their efforts toward continuous improvement and adjust their strategies in real-time based on ongoing data collection efforts. Data and feedback are needed at the classroom, program, school, and community levels to assess school change, all of which can be incorporated into the school's current improvement processes. In the end, the steps and pathways embedded in the CCM model seek to help schools in their efforts to undergo systematic changes that meet the academic and nonacademic needs of their youth, families, school, and community.

To date, research on the CCM model is emerging, yet most CCM studies independently examine the model with one school, district, or state initiative in the U.S. However, these studies have not been examined holistically to better understand the influence and successful predictors of implementing the CCM model in creating or assisting in school improvement efforts. To address this gap, the current study examined existing research on the CCM model using scoping review methods [35,36]. Unlike systematic reviews, scoping reviews aim to map key concepts that underpin a research area or topic. The value of scoping reviews is the examination of the research knowledge base, clarification of key concepts, and reports on the types of evidence that address and inform practice in the field [36]. Our scoping review was guided by two research questions: (a) What outcomes are associated with the CCM? and, (b) What are the barriers and facilitators to adopting and implementing the CCM?

**2. Methods**

The methodology developed by members of the Joanna Briggs Institute (JBI) [36] served as a guiding framework to ensure the quality and rigor of this scoping review. Using the definition put forth by JBI, our scoping review sought to map key concepts that underpin an emerging research area, with specific attention to time (e.g., 2006–2022), source (e.g., peer-reviewed articles), and origin (e.g., education-based studies in the U.S. where the model was created) [36]. Inclusion criteria were set around peer-reviewed articles that examined implementation practices, outputs, and outcomes associated with some or all components of the CCM model. More specifically, articles were included if they were published in English and after the model emerged in practice in 2006. Keywords in our search for articles included: "community collaboration model" and "expanded school improvement". Two researchers read each abstract and checked citations in each of these articles to ensure no additional studies were published or referenced and warranted inclusion in the current review. Exclusion criteria included removing articles that focused on expanded school improvement but did not articulate the adoption of the CCM or any of its specific implementation practices, or those that did not mention continual assessment of school-specific needs across the five CCM pathways. Furthermore, we excluded existing systematic or integrative review articles, dissertations, theses, books, book chapters, and other research outputs that were not published in peer-reviewed journals. Databases checked for relevant articles included PsycINFO and ERIC (EBSCO version). The literature search provided 14 peer-reviewed articles that met our inclusion criteria (see Figure 2). Two members of the research team then synthesized the methods, samples, and results of each article to identify outcomes associated with the CCM and to summarize implementation facilitators and barriers.

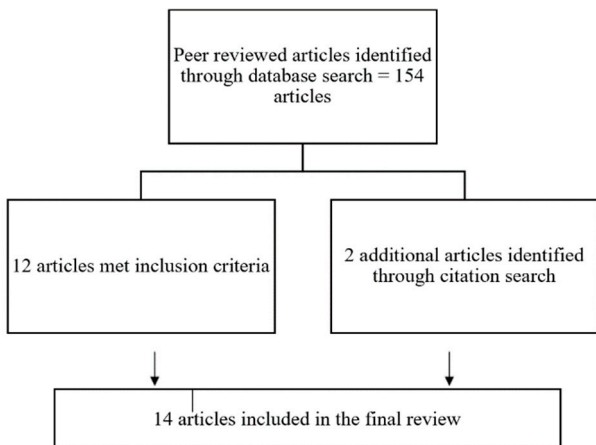

**Figure 2.** Summary of literature search process.

## 3. Results

After examining the literature, two areas related to the impacts of the CCM model were synthesized to answer our research questions. To answer our first research question, we synthesized findings from articles denoting the model's influence on student and school outcomes, along with the process and product innovations described in each study. In alignment with our second research question, our team examined facilitators and barriers identified in these studies as influencing the adoption and implementation of the CCM.

### 3.1. CCM Outcomes

3.1.1. Sample, Design, and Methodologies

Table 1 provides an overview of the nine articles examining outcomes associated with the CCM. Importantly, findings indicate that studies examining CCM outcomes utilized multiple designs and methodologies. For example, two studies used case study or qualitative designs and methodologies, such as interviews and focus groups, four were mixed methods designs, and three were quantitative designs exploring cross-sectional and longitudinal school climate perceptual data, behavioral indicators, and academic data. Furthermore, the samples across studies varied in size and included various school settings (i.e., elementary, middle, and high schools) and all studies were conducted in the United States. Two studies examined outcomes in rural schools, and three studies were conducted in urban school settings. Four studies took place in Title I schools designated as ones serving large percentages of students and families experiencing the effects of poverty. Our synthesis reflects outcomes associated with the CCM as reported in these studies that used diverse samples, designs, and methodologies.

**Table 1.** CCM outcomes.

| Authors | Sample | Research Design and Method | Student Outcomes | School Outcomes | Process and Product Innovations |
|---|---|---|---|---|---|
| Authors., et al. (2008) [37] | Five U.S. schools in one rural school district of approx. 2000 students | Case study | • Increased access to mental health providers | • Improved teacher/staff perceptions of support such as administrative guidance, mental health resources, and collaboration among school stakeholders<br>• Increased school-community partnerships | • Expanded funding Formed Mental Health Taskforce across three counties<br>• Established training on referral processes<br>• Created school improvement (S.I.) teams<br>• Developed data management infrastructure |

**Table 1.** *Cont.*

| Authors | Sample | Research Design and Method | Student Outcomes | School Outcomes | Process and Product Innovations |
|---|---|---|---|---|---|
| Authors., et al. (2010a) [28] | Three U.S. urban schools and three rural schools | Mixed methods (Secondary data, qualitative comments, and case examples) | • Improved academic performance (4th grade reading) | • Improved teacher/staff perceptions of student referral and support system<br>• Improved teacher/staff perceptions of resource availability<br>• Positive satisfaction (90%) indicators from families on service delivery of case management teams | • Designed new mental health partnerships<br>• Created multiple teaming structures<br>• Developed referral processes and assessment protocols<br>• Utilized data gathered to apply for grants<br>• Expanded funding (i.e., grants awarded)<br>• Changed roles and responsibilities of personnel to better align with needs<br>• Utilized data to inform decisions |
| Authors., et al. (2010b) [38] | Six U.S. schools (18-month implementation) and six districts (9-month implementation) | Qualitative (Documentation of school- and district-level capacity-related innovations and observation tools) | | • Increased awareness of nonacademic barriers to learning | • Expanded community partnerships<br>• Expanded use of multiple data sources<br>• Developed new programs<br>• Added personnel and supports in the schools<br>• Enhanced professional development<br>• Expanded funding<br>• Developed new policies and procedures |
| Authors., et al. (2016) [39] | Ten U.S. schools in one urban Midwestern school district (one high school, one middle school, three intermediate, and five elementary schools) | Mixed methods case study (Innovations checklist, CAYCI-SES data, academic performance indicators) | • Improved perceptions of school climate and academic motivation | • Improved from "academic watch" to "continuous improvement" designation in the state | • Enhanced professional development<br>• Created new policies and procedures<br>• Expanded funding<br>• Developed new school teams/systems<br>• Created new positions and roles<br>• Expanded use of multiple data sources<br>• Developed new partnerships, programs, services, and supports |
| Authors., et al. (2018) [40] | Four U.S. Title 1 elementary schools | Mixed methods (Records review, qualitative interviews and focus groups, secondary data) | • Improved academic performance<br>• Decreased rates of absenteeism and office of discipline referrals | • Improved teacher/staff perceptions of school climate<br>• Improved teacher/staff perceptions of the learning support system | • Created team structures<br>• Streamlined processes for referrals to CARE teams<br>• Developed partnerships focused on positive youth development (after-school programs, etc.) |
| Authors., et al. (2018) [41] | Four U.S. Title I schools (27 school professionals serving on CARE teams and 340 students) | Mixed methods (Qualitative interviews, pre- and post-data, secondary data on office of discipline referrals, absentee rates) | • Improved curriculum-based measures among students served by the CARE team<br>• Decreased office of discipline referrals among at-risk subgroup of students | • Improved interprofessional team collaboration through coordination of and access to services<br>• Improved follow-through on plans and interventions | • Created new referral processes<br>• Created regular communication processes and collaborative meetings among CARE team members |
| Authors., et al. (2020) [34] | One U.S. Title I elementary school in Midwest (132 students and 37 teachers/staff); Seven schools in one Intermountain West district (20,023 stakeholders) | Quantitative (Cross-sectional CAYCI-SES data) | • Improved perceptions of support for student learning<br>• Improved perceptions of academic press<br>• Decreased perceptions of externalizing behaviors | • Improved communication with parents/caregivers<br>• Improved teacher/staff perceptions of support for student learning | • Implemented teacher/staff professional development days<br>• Hired school-family-community coordinator<br>• Created with social-emotional workshops and parent/caregiver programs<br>• Implemented CARE teams |

**Table 1.** *Cont.*

| Authors | Sample | Research Design and Method | Student Outcomes | School Outcomes | Process and Product Innovations |
|---|---|---|---|---|---|
| Britt, N., et al. (2022) [42] | Two U.S. Title I urban elementary schools in a historically underserved neighborhood | Quantitative (Longitudinal data CAYCI-SES surveys, office of discipline referrals, absentee rates) | • Improved perceptions of safety and academic motivation<br>• Decreased office of discipline referrals over time<br>• Decreased rates of chronic absenteeism during virtual learning period (COVID-19) | • Improved communication across school stakeholders<br>• Improved teacher/staff perceptions of support for student learning | • Hired school-family-community coordinators in both schools<br>• Developed a coalition of over 20 partners committed to resource sharing<br>• Created multiple school teams<br>• Developed data tracking and monitoring systems<br>• Developed new programs/linkage to community supports (social skills program, afterschool programs, etc.) |
| Authors., et al. (2022) [43] | Five U.S. schools across one Intermountain West feeder pattern (over 5000 students) | Quantitative (Longitudinal data CAYCI-SES surveys, partnership documents, historic tracking of school activities/investments) | • Improved perceptions of belonging<br>• Improved perceptions of safety | • Improved parent/caregiver perceptions of school and community engagement | • Enhanced professional development opportunities for teachers/staff<br>• Implemented evidence-based interventions, programs, and processes (i.e., CARE teams)<br>• Added personnel and interventions in school (i.e., university interns) |

### 3.1.2. Student Outcomes

At the student level, our scoping review revealed five studies demonstrating how the CCM was associated with several positive student outcomes, such as increased access to mental health providers and improved student perceptions of school climate (i.e., academic motivation, support for student learning, academic press, belonging, and safety). Furthermore, four studies noted positive behavioral outcomes associated with the CCM, including decreased externalizing behaviors among students and fewer discipline referrals over time. Two studies reported declines in chronic absenteeism rates before and during the COVID-19 pandemic. Additionally, four articles revealed CCM implementation was associated with positive academic performance outcomes, including trends demonstrating improvement over time in literacy among students. Findings demonstrate the utility of the CCM in offering strategies to address nonacademic needs and systematically align priorities to improve school climate, student behaviors, and academic performance.

### 3.1.3. School Outcomes

Several school outcomes associated with CCM implementation were also identified in this scoping review, including ones such as enhanced stakeholder perceptions of the learning support system and improved positive indicators of school improvement based on state standards. Five studies found teachers and school staff reported improvements in their perceptions of nonacademic resources, academic learning supports, and student referral systems during CCM implementation. Teachers and school staff also benefited from the CCM, as evidenced by reports of behaviors denoting improved interprofessional collaboration (i.e., improved communication, follow-through, knowledge of resources, administrative guidance/support, etc.). Two studies also found that parent/caregiver perceptions of support and satisfaction with school services were associated with CCM implementation. Finally, one study signified several schools implementing the CCM moved from a state identifier of "academic watch" into "continuous improvement".

### 3.1.4. Process and Product Innovations

Our scoping review also identified two notable innovations: (1) Process innovations, i.e., how schools and their partners framed and performed joint work; and (2) Product

innovations, i.e., new operational structures and policies—for schools, community agencies, and their relationships. Examples of process innovations in all nine studies included strengthening systems by creating teaming structures, realigning roles and responsibilities of school personnel, and creating new positions (i.e., school-family-community coordinators), programs, partnerships, services, and supports. As reported in three studies, schools also benefited by having additional professional development opportunities and partnerships available to school stakeholders and expanded funding streams.

Examples of product innovations were described in seven studies. Team development was a priority and an outcome. Most studies indicated schools developed consultation, assessment, referral, and education teams (called "CARE Teams") to address nonacademic barriers to learning. Three studies also described improved data systems through the CCM implementation process, either via utilizing new data sources, strengthening the infrastructure for data management and analysis, or leveraging data to drive school improvement goals and priorities. Furthermore, two studies noted the CCM implementation process was associated with the development of new policies and practices.

*3.2. Barriers and Facilitators*

3.2.1. Sample, Design, and Methodologies

Table 2 synthesizes the nine studies that examined barriers and facilitators during the CCM implementation process. Comparable to findings related to CCM outcomes, studies exploring CCM implementation steps reflected diverse samples, designs, and methodologies. In total, five studies utilized case studies and qualitative designs, two used mixed methods, and three used descriptive quantitative designs using cross-sectional or longitudinal school climate, behavioral, and academic data. The samples across the included studies also reflected different school types and those from urban and rural districts, different geographic regions (i.e., Intermountain West and Midwest), and schools with students from diverse socioeconomic backgrounds (i.e., Title I school settings).

**Table 2.** CCM implementation barriers and facilitators.

| Authors | Sample | Research Design and Method | Barriers | Facilitators |
|---|---|---|---|---|
| Authors., et al. (2008a) [29] | Six U.S. schools and six districts (Twelve total sites) | Case study (Qualitative interviews and secondary CAYCI-SES data) | • Resistance among stakeholders<br>• Viewing the model as an "add on"<br>• Experiencing turnover among staff<br>• Limits to grant cycles/funding<br>• Lack of commitment to long-term change | • Strong leadership, especially from principals<br>• Embedding model within existing improvement structures<br>• Creating buy-in<br>• Access to consultants and liaisons<br>• Ability to adapt and tailor the model to meet needs |
| Authors., et al. (2008) [37] | Five U.S. schools in one rural school district of approx. 2000 students | Case study | • Reluctance to change systems and processes | • Developed collaborative leadership structures<br>• Engaged district leaders in the process<br>• Examined multiple sources of data<br>• Incorporated school stakeholders in the planning and refinement process<br>• Examined how to creatively blend and braid funding<br>• Redefined roles and responsibilities |

**Table 2.** *Cont.*

| Authors | Sample | Research Design and Method | Barriers | Facilitators |
|---|---|---|---|---|
| Authors., et al. (2010b) [38] | Six U.S. schools (18-month implementation) and six districts (9-month implementation) | Qualitative (Documentation of capacity-related innovations and observation tools) | • Competing local improvement agendas<br>• Lack of buy-in and support/personnel/time to engage in the planning process<br>• Viewing the process as an "add on"<br>• Experiencing turnover among staff<br>• Sociopolitical issues in the communities and schools· Sustaining funds/changing grant cycles | • Buy-in and knowledge of process among superintendents<br>• Strong leaders/central office administrators helping with the process<br>• Valuation among stakeholders of addressing non-academic barriers<br>• Embedding CCM into the existing planning process<br>• Tools and resources to help with implementation<br>• Access to consultants |
| Mendenhall, A., et al. (2013) [44] | Six U.S. schools (18-month implementation) | Qualitative case study (40 stakeholder interviews across six pilot settings) | • Lack of buy-in to the expanded school improvement approach<br>• Youths' complex academic and nonacademic barriers hindered participation in expanded supports<br>• Misunderstanding the model<br>• Time constraints Lack of funding for resources and programs | • Professional development supports to train stakeholders on processes and implementation needs<br>• Strong leadership at the district level<br>• Access to on-site consultants<br>• Readiness for change<br>• Collaboration among school and community stakeholders<br>• Communication with other schools involved in expanded school improvement efforts |
| Authors., et al. (2016) [39] | Ten U.S schools in one urban Midwestern school district (one high school, one middle school, three intermediate, and five elementary schools) | Mixed methods case study (Innovations checklist, secondary CAYCI-SES data, academic performance indicators) | • Time for joint decision-making procedures<br>• Challenges evaluating complex change initiatives<br>• Inability to capture perspectives from all stakeholders in the school | • University assistance with evaluation design<br>• Creation of new roles and responsibilities for staff<br>• Utilization of data for planning<br>• District leaders with high buy-in and interest |
| Authors., et al. (2018) [40] | Four U.S. Title 1 elementary schools | Mixed methods (Records review, qualitative interviews, and focus groups, secondary data on academic performance, office of discipline referrals, absentee rates, CAYCI-SES) | • Adding large programs and services quickly (i.e., start small and then grow)<br>• Lack of awareness among school stakeholders of student and school needs<br>• Turf wars among school professionals on roles and responsibilities<br>• Burnout among school and community stakeholders given high levels of need | • Having a clearly defined organizational structure through school teams<br>• Utilization of multiple data sources, including CAYCI-SES data<br>• Key leaders at the school and district levels guiding the work<br>• Obtaining buy-in from teachers and school leaders<br>• Offering professional development activities that provided practical tools for implementation in classrooms and throughout the school |
| Authors., et al. (2020) [34] | One U.S. Title I elementary school in Midwest (132 students and 37 teachers/staff); Seven schools in one Intermountain West district (20,023 stakeholders) | Quantitative (Cross-sectional CAYCI-SES data) | • Low buy-in and commitment during the implementation of CAYCI-SES from teachers/staff during Year 1 | • Increased buy-in from school administrators and teachers/staff during implement CAYCI-SES during Year 2 |
| Britt, N., et al. (2022) [42] | Two U.S. Title I urban elementary schools in a historically underserved neighborhood | Quantitative (Longitudinal data CAYCI-SES surveys, office of discipline referrals, absentee rates) | • Sustaining relationships and commitments over time with community partners | • Assistance from the local university in developing partnerships and analyzing CAYCI-SES data<br>• Utilization of data tracking and monitor system to drive decisions<br>• Creation of a coalition of community partners and school leaders to identify and share resources<br>• Partnership among university, local non-profits, school districts, and broader neighborhood revitalization efforts<br>• Partners sign MOUs to outline expectations of working together |

**Table 2.** *Cont*.

| Authors | Sample | Research Design and Method | Barriers | Facilitators |
|---------|--------|---------------------------|----------|--------------|
| Authors., et al. (2022) [43] | Five U.S. schools across one Intermountain West feeder pattern (over 5000 students) | Quantitative (Longitudinal data CAYCI-SES surveys, partnership documents, historic tracking of school activities/investments | • Limited time across partners for joint decision-making, planning, and problem-solving<br>• Lack of vision, misalignment of vision, and limited buy-in among leaders at districts, schools, and universities<br>• High turnover among faculty, staff, principals<br>• Limited funding to support university involvement, expertise, and research activities<br>• Challenges matching university research priorities to school needs | • Formalized partnerships with six universities/colleges<br>• Lead contact for community partnerships at the district level to coordinate services/supports<br>• Implementation of CAYCI-SES surveys annually |

### 3.2.2. Implementation Barriers

Nine studies included in the review explored barriers and facilitators to CCM adoption and implementation efforts. Key barriers identified across six studies included resistance, reluctance to change, viewing the process as an "add-on", and a lack of buy-in, commitment, and interprofessional collaboration among school stakeholders. Three studies also discussed internal and external challenges such as high turnover among school personnel, complexities during implementation based on the nonacademic needs of students, and trouble evaluating new practices and processes. Time was a barrier noted across three studies that negatively impacted progress toward key CCM milestones, including engaging in shared decision-making and building the school community. Additionally, four studies described funding limitations and sustainability issues that influenced the implementation of the CCM and the adoption of needed services, supports, or resources in schools.

### 3.2.3. Implementation Facilitators

Across the nine studies included in our scoping review, each article identified at least two facilitators that contributed to the successful implementation of the CCM. At the personnel level, eight studies noted buy-in and collaborative leadership structures (i.e., superintendents, principals, administrative or interprofessional teams, and district leaders) positively influenced the implementation of the CCM. The utilization of multiple forms of data to inform and guide implementation processes was the second most common facilitator identified and was described in six studies. Other facilitators identified across two or more studies included increasing access to consultants and liaisons to guide implementation milestones, integrating the model into existing processes and structures, and partnering with universities or other community entities to support targeted goals and priorities. Moreover, implementation facilitators included net, new resources; designing new roles and responsibilities; funding innovations (e.g., blending and braiding financial resources); outlining expectations for partners, and enhancing professional development activities for teachers and other school-based professionals.

## 4. Discussion

This scoping review sought to synthesize the state of the research to date exploring outcomes associated with the CCM and distilling facilitators and barriers to its adoption and implementation in schools. Our research team identified 14 peer-reviewed studies distilling both outcomes and implementation research on the CCM published between 2006 and 2022. Seven articles described outcomes and implementation factors associated with the CCM, bridging research on implementation processes and subsequent student- and school outcomes. Overall, the strengths of existing research on the CCM included examining implementation time-points, utilizing data from multiple school stakeholders and sources, and examining numerous implementation outputs. Outcomes and implementation factors also

were examined using several methods in varying geographic regions and demographically diverse schools and districts in the United States. The diversity of methods and samples across the included studies demonstrate the utility of model adoption for at-risk schools, especially in urban and rural settings. As a result, this scoping review fills an important gap in the literature by demonstrating how the CCM evaluates progress using multi-level school improvement indicators. Implementation and continuous improvement priorities also are identified. While the names for innovative initiatives focused on expanded school improvement vary, engagement of local community members in school change is a core idea guiding educational policies worldwide [14,15,26]. Based on the findings here, the CCM prioritizes the voice of the local community and their collective action to account for place-based poverty, social exclusion, and social isolation. The model's flexibility helps to contextualize local and specific needs, a process that is necessary given variability in state-specific educational policies in the U.S. and different global governing structures. Our results demonstrate how flexibility within the CCM allows expanded school improvement efforts to adapt and hold true to the statement that "place matters," a finding salient to research conducted in the U.S. and globally [5,6].

From an international perspective, the CCM may have utility given its ability to account for school location and community norms to guide evaluations and social analyses. No doubt adaptability is needed, as international comparative research exploring community school adoption and implementation across multiple eastern European and Eurasian countries points to need flexibility and variability needed due to policy contexts and leadership/governance structures [45]. Our findings also point toward the model cultivating a mutually beneficial, two-way relationship among educators and community constituencies, while community leaders commit to better, stronger schools. This phenomenon associated with the CCM, albeit studied only in the U.S. in our review, strengthens relationships across sectors and builds bridges amongst families, schools, and communities—priorities in educational contexts around the world. The CCM enables schools to draw on untapped family and community resources in support of learning and academic achievement, while families and community leaders gain access to school facilities and equipment, selected resources, and opportunities for expansive programs during out-of-school time.

Furthermore, this review demonstrates how school change occurs over time, cycles through stages, and leads to student- and school-level change. In alignment with our first objective, we found that the CCM is associated with improved perceptions of school climate amongst students in regard to their perceptions of support for learning, academic press, belonging, and safety. Students also demonstrated improved academic and behavioral outcomes and benefited from improved access to services. At the school level, perceptions of school climate improved over time among teachers and staff and parents and caregivers. Schools also demonstrated improvement indicators (i.e., movement to 'continuous improvement' status) and stronger processes and procedures as evidenced by improved referral systems, communication, and interprofessional collaboration. Importantly, stakeholders in schools also reported greater awareness of nonacademic needs and perceptions of resource availability in their schools. Several process and product innovations also emerged, including schools strengthening their teaming structures, utilizing multiple data sources, and identifying additional funding streams. Comparable to other expanded improvement models, the CCM extended and aligned school needs with resources beyond the school building (i.e., community, partner universities, parent coalitions, etc.).

A unique feature of the CCM model merits special emphasis. In contrast to partnership models that have minimal or no impact on school organizations, structures, and operational processes, the CCM prioritizes school improvement (indeed school reorganization) in tandem with community partnerships. Specifically, the CCM offers evidence-based, strategic opportunities to expand resources, services, and supports for students and teachers, providing links between community partners and school-based teams. What is more, CCM improvement priorities and pathways provide a mechanism for integrating and strengthening PYD. In this fundamental respect, the CCM aligns with other evidence-based

frameworks in the U.S., such as the coordinated school health model [16] and the Whole School, Whole Community, Whole Child model [19].

Beyond outcomes, our study also identified CCM implementation barriers and facilitators. Notably, when examining implementation processes, adoption is conceptualized as the change before a new approach begins [46]. Adoption barriers and facilitators identified in our review provide insight into how readiness is critical to successfully uptake the CCM over time. Our findings indicate school and district leaders, along with their community partners, need to work strategically before implementation to mitigate barriers identified in this review. For example, ensuring the model is integrated into existing structures is important during the implementation process. Identifying time to make collective decisions, outlining the responsibilities of key personnel, and examining competing school improvement agendas or priorities are essential to discuss before adopting the model. Addressing funding and sustainability concerns were also important and have been highlighted in other research on full-service community schools [14].

Our findings also demonstrate practices that helped increase buy-in and promote awareness of the CCM's ability to support improvement. For instance, in most studies included in this scoping review, perceptions of school climate improved for students, teachers/staff, and parents/caregivers. The findings were driven by the implementation practice of capturing perspectives from different stakeholders in schools in the United States. As schools look to adopt the CCM, leaders understanding the value of these data and using them to drive expanded efforts that extend beyond the school building and classroom may have a bidirectional and positive effect on the implementation process. Together, engaging in these practices as part of the CCM can systematically change how schools operate and address barriers to student learning, particularly as the CCM expands traditional reform efforts in alignment with school improvement planning processes and serves as more than an "add-on" social service delivery model.

### 4.1. Future Directions and Limitations

The CCM is distinct and also complementary to other partnership models that emphasize the importance of PYD, family support, and community development. Moreover, the CCM is dynamic and flexible offering specific steps to guide the implementation of this model. It also is one that can be adapted for local community contexts as well as public entities, starting with educational policy and ultimately including policies for mental health, health, and social welfare. Policy-focused research is an immediate priority, as well as translational research in education that helps to delineate linkages between policy and practice. The current scoping review has achieved these secondary aims in bridging elements of school improvement policies and tangible practices that guide and inform this work.

Drawing on the research synthesis offered in this review, CCM scholars and their practitioner partners have opportunities to build upon current findings and advance our understanding of the model as a strategy for expanded school improvement. Examining additional contextual factors beyond the United States and in coordination with different oversight bodies (i.e., commonwealth vs. state/local) can provide insight into the effectiveness of the CCM in other settings. Furthermore, examining implementation processes such as coaching, modeling, mentoring, leadership practices, and corrective feedback provided by consultants and administrators also can help further distill factors that could influence the successful adoption and implementation of the CCM. Utilizing systematic assessments (i.e., use of Australia's National School Improvement Assessment tool) to compare CCM outcomes and different social service delivery models or traditional school reform approaches are also needed to demonstrate the utility of the model.

It is important to note that the current review has several limitations. First, all studies were conducted in the United States, disallowing exploration of the utility of the model in other countries around the globe. The lead author also contributed to the development of the CCM and conducted a majority of the studies that met inclusion criteria. This means

studies included in our review are likely biased toward studies of efficacy (e.g., more ideal circumstances with researcher involvement) rather than effectiveness (e.g., translational research to practice). Our review also did not include a thorough search of grey literature, and despite following best practices in developing a scoping review and search strategy, it is possible that the search terms used did not exhaust all the available literature. Ultimately, opportunities to study the CCM more rigorously await researchers. For example, our review identified only a few studies that utilized longitudinal data to explore student outcomes over time and no studies examined outcomes in comparison schools. Quasi-experimental and randomized control designs would further strengthen the rigor of studies examining CCM outcomes and implementation processes, as well as help ensure identified changes are a direct result of the CCM and not a result of other forces or influences in the school setting.

One value added of the research, however, is that there has been a focus and exploration of school outcomes, an area often under-explored in research exploring school-family-community partnership models [20]. Scholars can also examine stakeholder perceptions across multiple groups (i.e., parents/caregivers, students, etc.). Often, studies examined perceptions of one or two stakeholder groups and those who respond to surveys may have held more favorable perceptions of needs and resources. Linking stakeholders' perceptional changes to CCM milestones and innovations also proved challenging. To examine how the CCM expands school improvement efforts, data from additional stakeholders would allow for a more in-depth exploration of perceptions of resources, partnerships, and system-level sustainability.

The shift from a conventional, stand-alone school improvement model to the CCM model (and comparable other models) includes important, companion priorities. For example, school professionals and their community counterparts need new work orientations and competencies, particularly those involving collaborative practices in substitution for solo roles and performances. The consequential roles of CCM-related change agents and professional developers are another priority. Processes and practices that help sustain CCM outcomes over time also are missing from the current research synthesis on this expanded school improvement model. What is more, few studies also examined how fiscal incentives, expanded funding, and collaboration grants the creation and sustainment of school-based or school-linked services. In doing so, school and district leaders can learn more about leveraging funding to increase and coordinate school mental health services to address youths' non-academic barriers. Researchers must continue exploring factors that help maintain school-family-community partnerships, secure commitment, and engagement in CCM milestones, and capture school change outcomes over time.

Finally, the comprehensive focus CCM necessitates new data systems, ones that bridge schools with community partners. In other words, holistic views of students depend on integrated, actionable data systems. Data "point persons" and specialists ultimately are a practical necessity. Last, but not least, CCM-related variability is inescapable, and this complicates research and research reviews. Alongside the generic features of the model, every study must attend to context- and time-specific factors and priorities. These inherent features of the CCM challenge researchers and evaluators. At the same time, the CCM emphasizes a practical necessity and a comparative advantage: This is not a "follow the numbers" or "a one-size-fits-all" approach. It is premised on the forthright acknowledgment of inherent variability among schools, communities, governments, and student and family populations. Future research on the CCM also could explore how certain sub-groups of students may benefit from targeted interventions within the five pathways of the model. Mixed method approaches to evaluation may be most suitable for this model to triangulate findings and monitor both implementation and outcomes associated with the model [14,47].

*4.2. Conclusions*

This scoping review sought to explore outcomes associated with the CCM and to identify barriers and facilitators influencing adoption and implementation. Fourteen articles utilizing diverse samples and different research designs illustrate the utility of the CCM as an evidence-based model of expanded school improvement. Across nine articles, the CCM was associated with positive student- and school outcomes. Key outcomes of adoption include increased teacher-staff commitment, improved perceptions of school climate, increased feeling of connectedness, increased support for basic needs, and increased access to community partners and services. Moreover, the high-quality research on implementation barriers and facilitators identified in nine studies provides a roadmap for schools and districts to replicate CCM milestones and evaluate their improvement progress using specific indicators, tools, and practices. Findings also can help them avoid roadblocks along the way. Research on the CCM demonstrates how contextual factors can aid or inhibit the adoption and implementation of expanded school improvements efforts and influence outcomes over time.

**Author Contributions:** Conceptualization, D.A.-B., S.B., H.A.L. and A.L.I.; methodology, S.B. and T.M.C.; validation, H.A.L.; formal analysis, S.B.; investigation, D.A.-B. and S.B.; writing—original draft, D.A.-B., S.B., H.A.L., T.M.C. and A.L.I.; writing—review and editing, D.A.-B., H.A.L., T.M.C. and A.L.I.; supervision, D.A.-B.; project administration, D.A.-B. All authors have read and agreed to the published version of the manuscript.

**Funding:** This research received no external funding.

**Institutional Review Board Statement:** Not applicable.

**Data Availability Statement:** Not applicable.

**Conflicts of Interest:** The authors declare no conflict of interest.

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
