# Peer review of "The Community Collaboration Model for School Improvement: A Scoping Review"

_education, doi:10.3390/educsci12120918_

Round 1

Reviewer 1 Report

The aim of the article is to examine existing research on the CCM model using scoping review methods. This is both appropriate and relevant, and the findings concerning the benefits of the CCM model are interesting.  In general, the article is well written and structured.

One problem, however, is to identify the “CCM model” in order to understand, what is being reviewed. Here I have a couple of minor points concerning the definition of CCM and the use of this definition in the analysis. The CCM model is presented in two definitions:

“Figure 1: The Community Collaboration Model for School Improvement”. For me it is too detailed to actually serve as an illustration of a theoretical or analytical point or as basis for a scoping review. But perhaps it is taken from practice in order to illustrate or document that this is how community collaboration is perceived in practice. If this is the case, it should be stated more explicitly.

The same goes for “Figure 1. CCM milestones” (please notice that they are both called figure 1). In total, the list includes 16 milestones. It is too detailed to being illustrative and/or to being examined in a generalized scoping review.

Consequently, the scoping review examines “some or all components of the CCM model”, and my guess is that most often, only some of the many component are included. Even though keywords in the search for articles included “community collaboration model” and “expanded school improvement”, it is hard to know whether these keywords covered the detailed definitions of CCM presented in the article.

In conclusion I suggest that these (and similar) concerns are discussed at the end of the article. I miss a section with self-critical discussions of methods and theories.

Author Response

Reviewer 1

The aim of the article is to examine existing research on the CCM model using scoping review methods. This is both appropriate and relevant, and the findings concerning the benefits of the CCM model are interesting.  In general, the article is well written and structured.

Response to reviewer: Thank you for this positive feedback concerning our article.

One problem, however, is to identify the “CCM model” in order to understand, what is being reviewed. Here I have a couple of minor points concerning the definition of CCM and the use of this definition in the analysis. The CCM model is presented in two definitions:

“Figure 1: The Community Collaboration Model for School Improvement”. For me it is too detailed to actually serve as an illustration of a theoretical or analytical point or as basis for a scoping review. But perhaps it is taken from practice in order to illustrate or document that this is how community collaboration is perceived in practice. If this is the case, it should be stated more explicitly.

Response to reviewer: Thank you for this feedback. The CCM is derived from practice and Figure 1 is meant to demonstrate to the reader how the efforts that guide the CCM are driven by school stakeholders and strategically mapped across 5 pathways. This is meant to show how the model contextualized that place matters as needs/gaps and priorities are localized based on needs/strengths in specific schools (i.e., areas for improvement). Albeit detailed, this logic model is meant to demonstrate how school stakeholder perceptions and needs inform where efforts are targeted, and then drive improvement across various intermediary- and long-term outcomes in partner schools. Previous research on CCM also uses this framework as the theory of change and has provided this specific figure for specificity.

The same goes for “Figure 1. CCM milestones” (please notice that they are both called figure 1). In total, the list includes 16 milestones. It is too detailed to being illustrative and/or to being examined in a generalized scoping review.

Response to the reviewer: Thank you for this feedback and important improvement to the paper. We agree that Figure 2 is too detailed and was already described in the text and is not needed as a graphic for the reader. Central to the model are the guided implementation steps that inform and drive the expanded school improvement process. We believe these implementation steps help to demonstrate the tangible actions hypothesized to contribute to school improvement outputs and outcomes examined in our scoping review. Given this, Figure 2 was omitted.

Consequently, the scoping review examines “some or all components of the CCM model”, and my guess is that most often, only some of the many components are included. Even though keywords in the search for articles included “community collaboration model” and “expanded school improvement”, it is hard to know whether these keywords covered the detailed definitions of CCM presented in the article.

Response to the reviewer: We went a step further as to say the articles included had to meet the definition of the CCM as defined when the model emerged in practice in 2006.

In conclusion I suggest that these (and similar) concerns are discussed at the end of the article. I miss a section with self-critical discussions of methods and theories.

Response to the reviewer: Thank you for this feedback and important point. We have included additional statements about the limitations of our study, including the potential to have missed grey literature and the lead authors role in the development of the CCM model.

Reviewer 2 Report

This study synthesized the research that explored outcomes associated with the Community Collaboration Model (CCM) using a scoping review method. The objective was to distill facilitators and barriers to its adoption and implementation in schools. Results show that the CCM can offer multi-level school improvement indicators to evaluate school-based progress. Implementation and continuous improvement priorities were identified.

While there is a valid argument for the much needed CCM, as it stands, the manuscript does not provide a strong introductory rationale preventing from making clear connection to the conclusion presented. For example, the introduction of the manuscript gave the sensation the study was about international issues, while the findings and conclusion seemed to focus on U.S. studies. The authors attempted to situate the study internationally, as school improvement, can be argued to be an international dilemma. However, the authors missed an opportunity to situate the study using U.S. issues (e.g., increased poverty rates via free and reduced price lunch).

Additionally, the introduction (and other parts in the conclusion) require additional evidence for the claims asserted. For instance, line number 59-64, the authors are making claims without addition relevant references. This is seen across the manuscript (e.g., lines 83-89). One important section that requires references is the Expanded Models of School Improvement (lines 142-156). The authors are building their case for models that go beyond the classroom without providing specific evidence that such models exist. They do self-cite using reference number 25, however, this is not enough to build their case. Their discussion section also is lacking reference to the research literature to compare their findings to such.

The authors argument for selecting a scoping review versus a systematic reviews requires additional rationale and evidence why their chosen method fits with their research question.

The discussion and conclusion are difficult to review, given that the introduction section speaks about international policies and issues, while the conclusion does not connect their presented argument.

While the authors attempt to make critical contribution to research by reviewing the CCM, as it stands, this manuscript requires major revisions before is fully considered for acceptance in this journal.

Author Response

Reviewer 2

This study synthesized the research that explored outcomes associated with the Community Collaboration Model (CCM) using a scoping review method. The objective was to distill facilitators and barriers to its adoption and implementation in schools. Results show that the CCM can offer multi-level school improvement indicators to evaluate school-based progress. Implementation and continuous improvement priorities were identified.

While there is a valid argument for the much-needed CCM, as it stands, the manuscript does not provide a strong introductory rationale preventing from making clear connection to the conclusion presented. For example, the introduction of the manuscript gave the sensation the study was about international issues, while the findings and conclusion seemed to focus on U.S. studies. The authors attempted to situate the study internationally, as school improvement, can be argued to be an international dilemma. However, the authors missed an opportunity to situate the study using U.S. issues (e.g., increased poverty rates via free and reduced-price lunch).

Response to the reviewer: Thank you for this comment. We framed the article from an international lens given the scope of the journal. In an attempt to keep the lens broad and also situate our study, we have included current mental health statistics and the number of students attending high-poverty schools in the United States and globally. Further, we emphasize how states rights also create variability in educational policy that reflect differences across countries on the global stage, and there is a need for adaptability (such as what the CCM affords) based on place mattering. References also are provided to support this claim.

Additionally, the introduction (and other parts in the conclusion) require additional evidence for the claims asserted. For instance, line number 59-64, the authors are making claims without addition relevant references. This is seen across the manuscript (e.g., lines 83-89). One important section that requires references is the Expanded Models of School Improvement (lines 142-156). The authors are building their case for models that go beyond the classroom without providing specific evidence that such models exist. They do self-cite using reference number 25, however, this is not enough to build their case. Their discussion section also is lacking reference to the research literature to compare their findings to such.

Response to the reviewer: We appreciate the reviewer’s comments in relation to providing supportive evidence. Relevant references were included to further ground claims made in these two areas of the paper outlined in the introduction and again emphasized in the discussion. Further, additional references are provided in the discussion to ground this study’s findings in other research findings related to expanded school improvement (see pages 590-614).

The authors argument for selecting a scoping review versus a systematic reviews requires additional rationale and evidence why their chosen method fits with their research question.

Response to the reviewer: Peters et al. created a standardized process to conduct scoping reviews. Authors mention a scoping review can help to examine a broad area of research to identify gaps or report on different types of evidence that address and inform the field of practice. This reference supports our argument for a scoping review whereby we examine both outcomes and facilitators and barriers to map the CCM based on time (e.g., since its emergence in practice in 2006), source (e.g., peer-reviewed articles), and origin (e.g., education-based contexts). Peters and colleagues argue scoping reviews include research questions, inclusion criteria, information about the types of participants, in our case schools, a design chart, and conclusions and implications for practice. The strengths of a scoping review in our study include seeking out credible evidence and mapping emergent outcomes and implementation practices without a rigid a priori protocol that answers a specific and delimited research question.

The discussion and conclusion are difficult to review, given that the introduction section speaks about international policies and issues, while the conclusion does not connect their presented argument.

Response to the reviewer: In our discussion we address the limitations of our study only focusing on studies in the U.S. but discuss how the CCM allows for flexibility and adaptability based on context; allowing findings to contribute to discussions of international school improvement policies and practices. We also included additional references to international studies calling attention to the need for adaptability and flexibility given different structures and processes across contexts.

Round 2

Reviewer 2 Report

Dear authors,

Thank you for addressing the changes suggested. After reviewing the revised manuscript, I am satisfied with the changes provided. If possible, one change that would make this an even stronger manuscript is contextualizing the study to the U.S. It is not until line 74 that you bring in the United States context. My concern is that the reviewed articles were conducted in the U.S. as you mentioned in the limitation section. Line 27 of the article notes three different countries, including the U.S. Could it be appropriate to reference the U.S. first and describe its initiatives and the subsequent sentence mention how it compares to the other countries?

I hope my suggestion is supportive to align the reviewed articles to the context set up at the introduction.

Author Response

Reviewer Comment:

After reviewing the revised manuscript, I am satisfied with the changes provided. If possible, one change that would make this an even stronger manuscript is contextualizing the study to the U.S. It is not until line 74 that you bring in the United States context. My concern is that the reviewed articles were conducted in the U.S. as you mentioned in the limitation section. Line 27 of the article notes three different countries, including the U.S. Could it be appropriate to reference the U.S. first and describe its initiatives and the subsequent sentence mention how it compares to the other countries? I hope my suggestion is supportive to align the reviewed articles to the context set up at the introduction.

Response:

Thank you very much for continuing to help us improve the manuscript by further contextualizing it to the U.S. in the introduction.  In response, new content is added in the first paragraph of the paper on Lines 26-45 (including the addition of an important citation further grounding the study in U.S. context) while also connecting the lens of school improvement to today’s policy context globally. We also inserted a few new clarifications throughout the manuscript to note the U.S. context specifically.